# A Statistical Method for Exploratory Data Analysis Based on 2D and 3D Area under Curve Diagrams: Parkinson’s Disease Investigation

**DOI:** 10.3390/s21144700

**Published:** 2021-07-09

**Authors:** Olga Sergeevna Sushkova, Alexei Alexandrovich Morozov, Alexandra Vasilievna Gabova, Alexei Vyacheslavovich Karabanov, Sergey Nikolaevich Illarioshkin

**Affiliations:** 1Kotel’nikov Institute of Radio Engineering and Electronics of RAS, Mokhovaya 11-7, 125009 Moscow, Russia; morozov@cplire.ru; 2Institute of Higher Nervous Activity and Neurophysiology of RAS, Butlerova 5A, 117485 Moscow, Russia; agabova@yandex.ru; 3FSBI “Research Center of Neurology”, Volokolamskoe Shosse 80, 125367 Moscow, Russia; doctor.karabanov@mail.ru (A.V.K.); snillario@gmail.com (S.N.I.)

**Keywords:** electromyogram, EMG, exploratory data analysis, wave train electrical activity analysis method, wave trains, wavelets, signal processing, AUC diagrams, ROC analysis, Parkinson’s disease

## Abstract

A statistical method for exploratory data analysis based on 2D and 3D area under curve (AUC) diagrams was developed. The method was designed to analyze electroencephalogram (EEG), electromyogram (EMG), and tremorogram data collected from patients with Parkinson’s disease. The idea of the method of wave train electrical activity analysis is that we consider the biomedical signal as a combination of the wave trains. The wave train is the increase in the power spectral density of the signal localized in time, frequency, and space. We detect the wave trains as the local maxima in the wavelet spectrograms. We do not consider wave trains as a special kind of signal. The wave train analysis method is different from standard signal analysis methods such as Fourier analysis and wavelet analysis in the following way. Existing methods for analyzing EEG, EMG, and tremor signals, such as wavelet analysis, focus on local time–frequency changes in the signal and therefore do not reveal the generalized properties of the signal. Other methods such as standard Fourier analysis ignore the local time–frequency changes in the characteristics of the signal and, consequently, lose a large amount of information that existed in the signal. The method of wave train electrical activity analysis resolves the contradiction between these two approaches because it addresses the generalized characteristics of the biomedical signal based on local time–frequency changes in the signal. We investigate the following wave train parameters: wave train central frequency, wave train maximal power spectral density, wave train duration in periods, and wave train bandwidth. We have developed special graphical diagrams, named AUC diagrams, to determine what wave trains are characteristic of neurodegenerative diseases. In this paper, we consider the following types of AUC diagrams: 2D and 3D diagrams. The technique of working with AUC diagrams is illustrated by examples of analysis of EMG in patients with Parkinson’s disease and healthy volunteers. It is demonstrated that new regularities useful for the high-accuracy diagnosis of Parkinson’s disease can be revealed using the method of analyzing the wave train electrical activity and AUC diagrams.

## 1. Introduction

The paper provides a detailed description of the method used for analyzing the wave train electrical activity in biomedical signals. The method was developed to investigate electroencephalograms (EEG), electromyograms (EMG), and accelerometer signals (tremorograms) in patients with Parkinson’s disease (PD) and identify regularities that are promising for the early diagnosis of this disease. Recently, many mathematical methods for analyzing EEG, EMG, and tremor signals have been developed. Historically, EMG analysis methods evolved from spectral analysis [1,2,3,4,5,6,7] and time-domain signal analysis methods such as morphological analysis [8], amplitude analysis [9], and autoregressive analysis [10,11] towards time–frequency domain analysis [12,13,14,15,16]. The state-of-the-art of EMG analysis methods is characterized by the active use of nonlinear data analysis methods [17], such as fractal analysis [18], phase analysis [19], recurrent quantification analysis [4,20,21], and the deep learning of neural networks [12,22,23,24,25]. According to the authors, the existing methods for analyzing EEG, EMG, and tremor signals, such as wavelet analysis [26,27,28], focus on local time–frequency changes in the signal and, therefore, do not reveal the generalized properties of the signal. By contrast, other methods, such as standard Fourier analysis, ignore local time–frequency changes in the signal and, therefore, lose a large amount of information that existed in the signal.

Let us consider the spectra of envelopes of EMG signals collected from PD patients and healthy volunteers (see Figure 1). On the left, an average spectrum of the tremor right hands of twelve PD patients and an average spectrum of the right hands of ten healthy volunteers are demonstrated. On the right, an average spectrum of the non-tremor left hands of the PD patients and an average spectrum of the left hands of the healthy volunteers are demonstrated. Three peaks are observed in the 4–10 Hz frequency range in the left figure. Two peaks are observed in this frequency range in the right figure. The Mann–Whitney statistical test discovers statistically significant differences in the spectra of the PD patients only in the tremor hands (see the left figure). These differences are well-known physiological regularity and are used for the diagnosis of PD [3]. However, the statistically significant differences between the spectra are not observed in the right figure. Therefore, the conventional spectral analysis does not reveal diagnostic features of PD in the non-tremor hands of the PD patients. In this paper, we will demonstrate that our method extracts much more information from the signals. In particular, statistically significant differences will be demonstrated between EMG signals collected from the non-tremor hands of the PD patients and healthy volunteers.

The wave train analysis method differs from standard signal analysis methods such as Fourier analysis and wavelet analysis in that it addresses the generalized characteristics of the biomedical signal based on local time–frequency changes in the signal. The idea of the method of wave train electrical activity analysis is to extract and analyze so-called wave trains in wavelet spectrograms. The wave train is the increase in the power spectral density (PSD) of the signal localized in time, frequency, and space. The wave trains correspond to local maxima in the wavelet spectrograms. We investigate the following wave train parameters: wave train central frequency, wave train maximal PSD, wave train duration in periods, and wave train bandwidth. Note that in [9,29,30,31,32,33], for EMG analysis, the term “burst” is used. However, the meaning of this term is different. Usually, the term “burst” refers to the EMG signal areas characterized by a sharp increase in amplitude. In contrast to these papers, we investigate the wave trains in the time–frequency domain but not in the time domain. We consider the biomedical signal as a combination of wave trains, and we do not consider the wave train as a special type of signal.

We extract the wave trains in a wide frequency range using complex Morlet wavelets. We consider local maxima in the wavelet spectrograms as wave trains. A technique based on so-called area under curve (AUC) diagrams is used to identify regularities in signals in a wide frequency range. The AUC diagrams are specially designed graphical diagrams that help to determine the wave train parameters that are characteristic of a given neurodegenerative disease. We distinguish the Frequency AUC diagram, Power spectral density AUC diagram, Duration AUC diagram, and Bandwidth AUC diagram. These types of AUC diagrams will be discussed in Section 2.4. Moreover, we distinguish 2D and 3D AUC diagrams. The 2D AUC diagrams are useful for manually searching regularities in the wave train electrical activity. The 3D AUC diagrams are used for searching statistically significant differences between groups of subjects using AUC diagrams. The AUC diagram technique is illustrated by examples of analyzing the EMG data from patients with Parkinson’s disease and healthy volunteers.

Note that the wavelets are not a critical issue of the method of wave train analysis. Generally speaking, similar data analysis can be carried out based on windowed Fourier transform. However, the wavelets have the following advantage: the time resolution of the wavelet changes automatically when different frequencies are investigated. Thus, the wavelets allow one to investigate wave trains simultaneously in high- and low-frequency bands. We use the Morlet wavelet because it is simple and people can easily understand the wavelet diagrams. Our method differs from other methods based on wavelets [15,16] in that the wave trains are considered and AUC diagrams are applied.

The problem of the early and differential diagnosis of PD is all too real [17,34,35,36,37]. It is difficult to identify the early features of PD because the disease develops over a long time without clear clinical manifestations. The first clinical stage of PD is characterized by the patient having a pathological tremor on only one side of the body. At the same time, another side of the body does not demonstrate the clinical manifestations of PD (has no trembling hyperkinesis [38,39]). O. E. Khutorskaya [1,2] suggested that the non-tremor side of the body of PD patients can be considered as a model of the preclinical (early) stage of PD. Therefore, it is important to investigate the non-tremor side of the body of first-stage PD patients. This paper demonstrates that wave train analysis can reveal new regularities in the non-tremor side of the PD patient body which are useful for the diagnosis of Parkinson’s disease at the preclinical stage.

The method used for analyzing the wave train electrical activity of signals is discussed in Section 2. Section 3 describes the results of the group data analysis. A discussion of the data analysis results is given in Section 4.

## 2. Materials and Methods

The wave train is the increase in the signal PSD localized in space, time, and frequency. We applied wavelet spectrograms calculated using the complex Morlet wavelet to determine wave trains in signals. An adaptive two-dimensional Gaussian filter was used to smooth the wavelet spectrogram to eliminate artifacts arising in the process of calculating wavelets. Then, we detected the local maxima on the wavelet spectrogram. The attributes of the wave trains were calculated, such as the central frequency of the wave train, the maximal PSD of the wave train, the duration of the wave train in periods, and the bandwidth of the wave train.

### 2.1. Experimental Data

The object of our investigation is the electromyographic signals in PD patients at the first stage of the disease according to the classical Hoehn–Yahr scale. Approximately half of the patients had never taken antiparkinsonian drugs before, and the other patients had not taken antiparkinsonian drugs for one to two days before the investigation. Additionally, a group of healthy volunteers participated in the investigation. The average age of the patients was 56 years (the minimum age was 38 years; the maximum age was 69 years). The average age of the healthy volunteers was 51 years (the minimum age was 24 years; the maximum age was 71 years). There were no statistically significant differences between the ages of the patients and the healthy volunteers (the Mann–Whitney test was used). Note that the group of PD patients included patients with left-hand tremor (10 persons) and patients with right-hand tremor (12 persons), with 22 persons in total (see Figure 2). All the patients were examined at the FSBI Research Center of Neurology, and PD was diagnosed. The number of healthy volunteers was 10 persons. All patients and healthy volunteers were right-handed.

The subjects were sitting in a chair during the data acquisition. Arms were outstretched forward. The duration of the single recording was 1 min and 30 s. EMG electrodes were placed on both arms of the patient on the antagonist muscles of the wrist joint (extensor and flexor muscles: Musculus extensor carpi radialis longus and Musculus flexor carpi radialis). The eyes were closed during the measurement. The Neuron-Spectrum-5 multifunctional system for neurophysiological studies (Neurosoft Ltd.) was used for EMG recording. The sampling rate was 500 Hz. The Butterworth high-pass filter with a cut-off frequency 0.5 Hz and a 50 Hz notch filter were used during the data acquisition.

### 2.2. Signal Preprocessing

The preprocessing of EMG signals included the following stages:The 50, 100, 150, and 200 Hz notch filters removed the power line interference.The 60–240 Hz fourth-order Butterworth bandpass filter was applied to EMG in the forward and reverse directions.The envelope of the EMG signal was calculated using the Hilbert transform. The signal envelope was used for tremor analysis according to the classical method [1,2].The envelope of the signal was decimated; the decimation factor was 4.

### 2.3. Calculation of Local Maxima in the Wavelet Spectrogram

We used the wavelet spectrograms calculated using the complex Morlet wavelet (Equation 1) to determine wave trains in the signals:(1)$$\psi \left(x\right)=\frac{1}{\sqrt{\pi F_{b}}}exp\left(2\pi ıF_{c}x\right)exp\left(\frac{-x^{2}}{F_{b}}\right)$$
where $F_{b}$ = 1 and $F_{c}$ = 1. We calculated the wavelet spectrogram in the frequency range from 0.1 to 50 Hz in the examples considered in the paper; the frequency step was 0.1 Hz.

The wavelet spectrogram was smoothed by an adaptive two-dimensional Gaussian filter to eliminate artifacts arising in the process of calculating the wavelets. The width of the Gaussian window in time and frequency depends on the width of the time and frequency windows of the wavelet at the considered frequency. We used a smoothing window width that was twice less than the time and frequency width of the wavelet window. The width of the smoothing window should be less than the width of the wavelet window to prevent the distortion of the wavelet spectrogram shape.

Let us consider an example of a wave train on a wavelet spectrogram of EMG signal in an extensor muscle of the non-tremor (right) arm of a patient with the left-side tremor of the body (Figure 3). The central frequency of the wave train is 15.2 Hz; the signal is clearly distinguished in the time–frequency space.

The envelope of the EMG signal (Figure 3) is demonstrated in Figure 4 (on the left). One can see three periods of the wave train envelope in the figure. On the right, the source EMG signal is demonstrated. It is almost impossible to reveal the wave train considered in the source signal without special processing. Therefore, the standard methods for the morphological analysis [8] of signals are inapplicable for this signal.

In Figure 5, other examples of the wave trains are demonstrated. On the left, the envelope of the EMG signal in the tremor left hand of a PD patient is demonstrated. On the right, the envelope of the EMG signal in the left hand of a healthy volunteer is demonstrated. The wave trains are very similar. The central frequency of both wave trains is ~6.5 Hz. In the framework of our method, we do not try to distinguish “normal” and “abnormal” wave trains. Instead, we use a statistical analysis based on the number of detected wave trains.

Note that the computation of wavelet spectrograms and detection of wave trains are the most time-consuming data processing steps. The processing of the EMG data for the total group of subjects (32 persons) takes about 2 hours on a 2.30 GHz PC machine. We do not consider the wave trains in the wavelet spectrogram if the duration of the wave train is less than 1/10 of the signal period at the central frequency of the local maximum to increase the speed of computation.

The time duration and the frequency width of the local maximum are measured at the $1/\sqrt{2}$ maximum height of the local maximum. Figure 6 demonstrates time and frequency slices of the wave train wavelet spectrogram.

The wave train can be characterized by several parameters: the leading (central) frequency, the maximal PSD, the duration in periods (at $1/\sqrt{2}$ maximum height), and the bandwidth (at $1/\sqrt{2}$ maximum height). These parameters form a multidimensional space. The analysis aims to select a certain subspace in the given space where a difference between the groups of subjects is observed. The following notation is used below to denote the subspace bounds: *MinFreq* (the minimal wave train frequency), *MaxFreq* (the maximal wave train frequency), *MinPSD* (the minimal wave train PSD), *MaxPSD* (the maximal wave train PSD), *MinDurat* (the minimal wave train duration in periods), *MaxDurat* (the maximal wave train duration in periods), *MinBandwidth* (the minimal wave train bandwidth), and *MaxBandwidth* (the maximal wave train bandwidth).

The number of wave trains in the PD patients was compared with the number of wave trains in the healthy volunteers using ROC curves. The quality of the ROC curve is characterized by the area under the ROC curve (AUC). AUC values from 0 to 1 can be obtained when comparing groups of subjects. We were interested in values that significantly differed from 0.5—that is, values close to 0 and 1. These AUC values have the following interpretation. AUC > 0.5 means that the number of wave trains is higher in the patients than in the healthy volunteers. AUC < 0.5 means that the number of the wave trains is higher in the healthy volunteers. Both cases are of interest for the investigation and the diagnosis of PD.

We developed so-called AUC diagrams to search for regularities in the multidimensional space of the wave train parameters. We considered the following types of AUC diagrams: 2D and 3D diagrams.

### 2.4. 2D AUC Diagrams

The 2D AUC diagram demonstrates the AUC values corresponding to different ranges of the given wave train parameter. The range of the wave train parameter is characterized by the lower and upper bounds—namely, the minimal and maximal values of the parameter. The abscissa indicates the lower bound of the considered parameter, while the ordinate indicates the upper bound of the considered parameter. The AUC value is displayed using a colormap. The standard jet colormap is applied in the examples given in this paper. Low AUC values are displayed in blue and high AUC values are displayed in red in this colormap. AUC values close to 0.5 are displayed in green.

We considered AUC diagrams of different types—namely, Frequency AUC diagrams (see example in Figure 7), Power spectral density AUC diagrams (see example in Figure 8), Duration AUC diagrams (see example in Figure 9), and Bandwidth AUC diagrams (see example in Figure 10).

Let us consider the frequency range from 1 to 50 Hz and calculate the number of wave trains in the EMG signals of each PD patient and each healthy volunteer. The patients with the left-hand tremor and the patients with the right-hand tremor were investigated separately.

An example of a Frequency AUC diagram demonstrates the AUC values calculated for various ranges in the frequency interval from 1 to 50 Hz with 1 Hz steps (see Figure 7). Corresponding ROC curves compare the number of wave trains in the extensor muscle in the right non-tremor arm of the left-hand-tremor PD patients with the number of wave trains in the extensor muscle in the right arm of the healthy volunteers. The red color in the diagram indicates that the number of wave trains in the patients is greater than that in the healthy subjects. The blue color in the AUC diagram indicates that the number of the wave trains in the patients is lower than the number in the healthy subjects. The diagram has a triangular shape because the upper bound of the range is always bigger than the lower bound of the range.

Reading the diagram should be carried out in the following way. One should start by looking at the AUC values located on the diagonal line of the diagram. In the Frequency AUC diagram, the diagonal line corresponds to narrow frequency ranges MinFreq≈MaxFreq, which allows one to accurately estimate the frequencies where differences appear between the patient group and the control group. These frequencies correspond to red and blue dots on the diagonal line. Next, one should consider the monochromatic areas adjacent to the diagonal line. The area must be of the same color as the red/blue dot on the diagonal line. The bigger the area is, the stronger the revealed difference between the groups of subjects is.

In Figure 7, two bright red areas are observed in the frequency range. The first red area is situated along the abscissa axis from 0 to 18 Hz; the y-coordinate is equal to approximately 20 Hz. The second area is situated along the ordinate axis from 17 Hz and above; the x-coordinate is equal to approximately 14 Hz. The brightest point has coordinates of 8 Hz on the abscissa axis and 20 Hz on the ordinate axis. This point corresponds to the frequency range from 8 to 20 Hz. The red color indicates that the PD patients have more wave trains than the healthy subjects in the human physiological tremor frequency area. The AUC value is approximately 0.88 in this frequency range; thus, the observed regularity can be used as a diagnostic criterion for Parkinson’s disease.

Let us consider a Power spectral density AUC diagram (Figure 8). The diagram is calculated on the same dataset. In contrast to the Frequency AUC diagram, the ranges of PSD are considered in the Power spectral density AUC diagram. The range of PSD is characterized by the lower and upper bounds. The abscissa indicates the lower bound of the PSD range, while the ordinate indicates the upper bound of the PSD range. The values of the wave train PSD are considered in the interval from 0 to 1000 μV${}^{2}/$ Hz with 10 μV${}^{2}/$ Hz steps. In the figure, a bright red area is observed along the ordinate axis above 70 μV${}^{2}/$ Hz.

The third type of AUC diagrams is the Duration AUC diagram. Let us consider an example of the Duration AUC diagram (see Figure 9). The diagram is based on the same dataset as the previous diagrams; however, the ranges of wave train durations are considered. As in previous figures, the duration range is characterized by the lower and upper bounds of the range. The abscissa indicates the lower bound of the range, while the ordinate indicates the upper bound of the range. The duration of the wave trains is considered in the interval from 0 to 10 periods with 0.1 period steps.

Figure 9 demonstrates a bright yellow area with the x-coordinate of less than 3.8 periods. A narrow bright orange area is situated along the ordinate axis; the x-coordinate is equal to approximately 3.8 periods. This diagram can be explained in the following way: most wave trains have a duration of approximately 3.8 periods, but shorter wave trains are observed too.

Let us consider an example of a Bandwidth AUC diagram (Figure 10). The diagram is based on the same dataset. In the Bandwidth AUC diagram, the ranges of bandwidth of the wave trains are considered. The bandwidth range is characterized by the lower and upper range bounds. The abscissa axis indicates the lower bound of the range, while the ordinate axis indicates the upper bound of the range. In this example, the frequency bands in the interval from 0 to 50 Hz are considered; the step size is 0.1 Hz.

The frequency bandwidth of the signal characterizes the shape of the signal. The narrowband signal is close to the harmonic one; the wideband signal contains fragments of a complex shape. Figure 10 demonstrates several areas corresponding to multidirectional differences in the wave train bandwidth between the groups of subjects. In particular, a bright orange area is observed along the abscissa axis from 0 to 7 Hz; the y-coordinate is equal to approximately 7 Hz. In addition, a light blue area is observed along the abscissa axis from 0 to 4.5 Hz; the y-coordinate is equal to approximately 4.3 Hz. There are also vertical orange columns with x-coordinates equal to approximately 17 Hz and 26 Hz. Note that multidirectional effect diagrams are more difficult to interpret. In this example, we can only conclude that it is possible to obtain and investigate multidirectional differences between the groups of subjects by detailing the wave train bandwidth ranges. This will be carried out in the further steps of analysis.

The analysis of the wave train electrical activity begins with the calculation of AUC diagrams of all four types (see Figure 7, Figure 8, Figure 9 and Figure 10). At the first stage of the analysis, one has to choose one out of four diagrams that demonstrates the most pronounced differences between the patients and healthy volunteers—that is, the diagram that contains the most prominent red or blue area with AUC values close to 0 or 1. The selected red/blue area corresponds to a certain range of the corresponding parameter. The calculation of all four diagrams is repeated in the next steps of the analysis. However, only the wave trains that correspond to the ranges of the wave train parameters selected in the previous steps are taken into account. It is possible to identify interesting ranges of all four parameters of the wave trains (the central frequency, PSD, duration in periods, and bandwidth) by iteratively repeating the described operations (see the flowchart in Figure 11).

In the Frequency AUC diagram (Figure 7), we chose a red area with the following coordinates: frequency range from 8 to 20 Hz. Therefore, we will consider only the wave trains that belong to the frequency interval from 8 to 20 Hz in the further steps of the analysis. Let us recalculate the other three diagrams taking into account the chosen constraint.

Figure 12 demonstrates the Power spectral density AUC diagram for the wave trains that belong to the frequency interval from 8 to 20 Hz. The diagram differs from Figure 8 because the frequency band of the considered wave trains is narrowed. The colored areas in Figure 12 are brighter, but the size and position of the areas are approximately the same. This means that the applied frequency constraint allows the better recognition of the differences between the groups of subjects. The diagram demonstrates the most substantial differences between the patients and control subjects in the following point: the PSD range from 30 to 700 μV${}^{2}/$ Hz (AUC = 0.96). The maximal PSD of the wave trains may be a hardware-dependent characteristic; thus, one can consider only the wave trains with a PSD above 30 μV${}^{2}/$ Hz in the further steps of the analysis.

Let us consider the Duration AUC diagram for the wave trains that belong to the frequency interval from 8 to 20 Hz (see Figure 13). The diagram differs from Figure 9 because the frequency band is narrowed. The colored areas on the diagram became brighter. A red column appears along the ordinate; the x-coordinate is equal to approximately 1 period. This means that one can better distinguish the groups of subjects when the frequency band of the wave trains is narrowed. The most substantial differences between the patients and control subjects are observed in the wave train duration range from 0.7 to 2.6 periods (AUC = 0.88). Thus, we can narrow the interval of the wave train durations in the further step of the analysis. Only durations from 0.7 to 2.6 periods will be considered.

Figure 14 demonstrates the Bandwidth AUC diagram. In the diagram, the wave trains belong to the frequency interval from 8 to 20 Hz. The diagram differs from Figure 10 because the frequency band is narrowed. The size and position of the red column have changed. A bright red area appears along the ordinate axis; the x-coordinate is equal to approximately 2 Hz. This means that the bandwidth of the wave trains characterizing PD differs sufficiently from the bandwidth of the other wave trains observed during the medical examination.

Now we are ready to implement the next iteration of the analysis. Once again, we have to choose which one of the four diagrams demonstrates the most striking regularities. A new constraint is applied to the wave train parameters based on this diagram. Let us choose the Power spectral density AUC diagram at this stage. We apply the following constraint based on this diagram: PSD no less than 30 μV${}^{2}/$ Hz. Let us recalculate the other three AUC diagrams (frequency, duration, and bandwidth), taking into account two constraints: a frequency from 8 to 20 Hz and a PSD no less than 30 μV${}^{2}/$ Hz.

Figure 15 demonstrates the Frequency AUC diagram with a PSD that is no less than 30 μV${}^{2}/$ Hz. Note that the frequency constraint needs to be refined using this diagram. The Frequency AUC diagram (see Figure 15) has changed substantially in comparison with that of Figure 7. The red areas in Figure 15 are brighter and larger. The blue areas have disappeared. Thus, the applied constraints made the differences between the groups of subjects more contrasting. The diagram demonstrates the strongest differences between the patients and control subjects in the frequency range from 8 to 20 Hz; the AUC value is equal to 0.93. Note that the disappearing blue area may correspond to another statistical regularity that differentiates the groups of subjects; however, this regularity is out of the scope of this paper.

In Figure 16, the Duration AUC diagram is demonstrated. In this diagram, the following constraints are applied to the wave train parameters: a frequency from 8 to 20 Hz and a PSD no less than 30 μV${}^{2}/$ Hz. Figure 16 differs slightly from Figure 13. The red areas on the diagram became more intense. An intense red area appears inside the red region. The intense red area has the following coordinates: x-coordinates from 0 to 0.6 periods and y-coordinates from 3.6 periods and more. This diagram can be interpreted in the following way. The duration of most wave trains typical for the PD patients is equal to approximately 1 period. However, there are shorter and longer wave trains as well. Thus, the better recognition of PD patients is obtained when considering the wave trains in a wider range from 0.5 to 4 periods. The AUC value in the detected intense red area reaches 0.93, which is sufficient for the high-quality recognition of PD patients.

Figure 17 demonstrates the Bandwidth AUC diagram for wave trains that have a frequency from 8 to 20 Hz and a PSD no less than 30 μV${}^{2}/$ Hz. The diagram differs from Figure 14 because of the constraints applied to the wave train parameters. The red areas on the diagram became more intense, while the multidirectional effects are still observed as the red and blue areas present on the diagram. This means that we cannot confidently determine the characteristic bandwidth of the wave trains typical for PD based on the available dataset. We can only conclude that the frequency bandwidth of the wave trains belongs to a wide interval; the value of the bandwidth can rise to ~28 Hz.

At the current step of the analysis, further iterations of the analysis do not change the Frequency AUC diagram (Figure 15), Power spectral density AUC diagram (Figure 12), Durations AUC diagram (Figure 16), and Bandwidth AUC diagram (Figure 17). Further detailing of the duration and bandwidth of the wave trains does not improve the AUC values. Thus, the iterative process of fitting the wave train characteristics typical for PD patients can be completed at this point. It was determined that the wave trains that help to distinguish the PD patients and healthy subjects have the following attributes: a frequency band from 8 to 20 Hz, a maximum PSD no less than 30 μV${}^{2}/$ Hz, a duration from 0.5 to 4 periods, and a bandwidth from 1 to 28 Hz. The Mann–Whitney test confirms that a statistically significant difference between the numbers of wave trains with the given attributes is observed in PD patients and healthy volunteers (*p* ≤ 0.0011).

In the general case, the iterative process of refinement of the wave train parameters is to be completed in the following situations:Any further restrictions on the parameters of the wave trains do not change the AUC diagrams. This means that the further refinement of the wave train parameters makes no sense.The refinement of the wave train parameters worsens the AUC values sufficiently in the AUC diagrams. This means that the investigated ranges of the wave train parameters became too narrowed; the number of wave trains considered in the AUC diagrams is too small. Theoretically speaking, in this case the refinement of the wave train parameters could be continued. However, the available dataset is not sufficient for this. The investigation of the wave train parameters could be continued if the number of subjects and/or the duration of EMG records are sufficiently increased.

Note that the initial AUC diagrams (Figure 7, Figure 8, Figure 9 and Figure 10) had several red and blue areas that serve as a starting point for the iterative refinement of the wave train parameters. In the considered example, only one wave train type observed in the right non-tremor arm of the PD patients with a tremor in the left arm was investigated. The results of the analysis of other regularities observed in the dataset are given in Table 1 and Table 2. In particular, we analyzed the EMG signals in the left arm of the left-hand-tremor PD patients, the EMG signals in the left non-tremor arm of the right-hand-tremor PD patients, and the EMG signals in the right arm of the right-hand-tremor PD patients. Thus, we analyzed both the non-tremor arms and arms with trembling hyperkinesis of the PD patients. Table 1 contains the results of the iterative analysis of the wave train parameters in the extensor muscles. Table 2 contains the results of the iterative analysis of the wave train parameters in the flexor muscles.

### 2.5. 3D AUC Diagrams

Wave train electrical activity analysis based on 2D AUC diagrams requires considering a large number of combinations of the upper and lower bounds of parameter ranges. It is useful to check what positions of the 2D AUC diagram correspond to statistically significant differences between the numbers of wave trains in the groups of subjects compared. We implement this check using the Mann–Whitney nonparametric test; however, the multiple comparisons problem arises. The essence of the multiple comparisons problem is that the statistical test may give a sufficient first-type error when a large number of ranges of parameters are checked in the 2D AUC diagram. The simplest way to solve the multiple comparisons problem is to apply Bonferroni correction [40]. The value of the Bonferroni correction depends on the number of cells in the 2D AUC diagram. We call the number of rows/columns in the 2D AUC diagram the resolution *R* of the diagram. If the resolution of the AUC diagram is high, the value of the Bonferroni correction also becomes high. The Bonferroni correction changes (Equation 2) the alpha level of the Mann–Whitney test using the correction coefficient *C*:(2)$$\alpha {}_{B}=1-{\left(1-\alpha {}_{0}\right)}^{1/C}$$
where $\alpha {}_{0}$ = 0.05. Let the *C* correction coefficient (Equation 3) be equal to the number of cells in the upper triangle of the 2D AUC diagram, including the number of cells on the diagonal of the AUC diagram:(3)C=R(R+1)/2

The value of the correction coefficient depends on the *R* resolution according to the quadratic law. Therefore, the probability of detecting statistically significant differences in the 2D AUC diagram sufficiently decreases when the resolution is high. On the other hand, if the resolution *R* is low, the detailing of the 2D AUC diagram is reduced, and one can miss certain regularities present in the dataset. Thus, it is necessary to find a compromise between the level of detail in the 2D AUC diagram and the value of the Bonferroni correction to reveal interesting statistically significant differences between the groups of subjects. We developed a special type of AUC diagram, named a 3D AUC diagram, to solve this problem. The 3D AUC diagram is a generalization of the 2D AUC diagram. The abscissa and ordinate axes on the 3D AUC diagram indicate the values of the upper and lower bounds of the range of the considered wave train parameter, as in the 2D AUC diagram. However, the applicate axis indicates the *R* resolution of the AUC diagram. Thus, the horizontal slice of the 3D AUC diagram is a case of the 2D AUC diagram. In the 3D AUC diagrams, only the points that correspond to statistically significant differences between the numbers of wave trains in the groups of subjects are displayed; Bonferroni correction (Equation 2), which depends on the *R* resolution, is taken into account when the statistical significance is checked.

Let us consider an example of the 3D AUC diagram (see Figure 18). This diagram is a form of Frequency 3D AUC diagram; it demonstrates statistically significant differences between the numbers of wave trains in the groups of subjects when considering various ranges of frequencies. The number of wave trains in the left arm of the left-hand-tremor PD patients and the number of wave trains in the left arm of the healthy subjects are compared. The values of the wave train parameter ranges are given in Table 1, line 5. The 3D AUC diagram demonstrates a 3D isosurface that corresponds to various $p\leq \alpha {}_{B}$. The upper plateau of the isosurface corresponds to the resolution of 23. The coordinates of the plateau are from 3.1 to 3.4 Hz along the abscissa axis and from 6.4 to 7.5 Hz along the ordinate axis. The horizontal slice area of the isosurface grows and then decreases when the resolution decreases. This is because the Bonferroni correction is softened; however, the degree of detail in the diagram also decreases. The top point of the isosurface is of interest because it reveals statistically significant differences in the dataset with the greatest degree of detail. In the example being considered, the 3D AUC diagram confirms that there are statistically significant differences between the numbers of wave trains in the groups of subjects in the frequency range from about 3 to 7 Hz.

Figure 19 demonstrates the Frequency 2D AUC diagram that corresponds to the horizontal slice of the 3D AUC diagram (Figure 18) at the resolution *R* = 15. Figure 19 includes only the points that correspond to statistically significant differences between the groups of subjects. The diagram demonstrates that the slice has an irregular shape with coordinates from 2.28 to 4.2 Hz along the abscissa axis and from 6.12 to 8 Hz along the ordinate axis. The AUC values in the 2D AUC diagram are higher than 0.98; there is an area with high AUC values up to 1 in the central part of the diagram. The coordinates of this area are from 2.3 to 4 Hz along the abscissa axis and from 6.12 to 8 Hz along the ordinate axis. This example demonstrates that one can obtain the best AUC values when choosing the optimal level of detail in the AUC diagram; this allows high accuracy when diagnosing PD.

## 3. Group Data Analysis

The analysis of 2D and 3D AUC diagrams that compare PD patients and healthy volunteers revealed several types of wave train electrical activity that are a distinctive feature of PD patients. Let us consider some regularities discovered in the dataset to clarify what neurophysiological mechanisms may control these types of electrical activity.

Let us consider the scatter plot (see Figure 20) that demonstrates the number of wave trains detected in two frequency intervals in the extensor muscle of the arms with trembling hyperkinesis in the PD patients: the physiological tremor frequency interval and the Parkinsonian tremor frequency interval. The abscissa axis corresponds to lines 3 and 4 in Table 1. The ordinate axis corresponds to lines 5 and 6 in Table 1. We have included the characteristics of the healthy subjects in the scatter plot for comparison. Each point in the scatter plot corresponds to one subject. The PD patients are indicated by the red color; the control subjects are indicated by the green color.

The point cloud corresponding to the PD patients has an elongated shape (see Figure 20). The point cloud corresponding to the healthy subjects is situated in the lower right corner of the scatter plot. Note that the point clouds can be easily separated. The PD patient point cloud is perpendicular to the diagonal of the scatter plot, which is evidence of the negative correlation between the wave train numbers corresponding to physiological and Parkinsonian tremors.

The check of the correlation confirmed that the correlation between the number of wave trains corresponding to the physiological and Parkinsonian tremors is statistically significant in the right-hand-tremor PD patients (see Figure 20, on the right); the correlation coefficient is −0.6885, the first-type error probability is 0.0133, the Spearman’s rank correlation coefficient is −0.7483, and the first-type error probability in the Spearman’s nonparametric test is 0.0074.

The investigation of these types of wave trains in the left-hand-tremor PD patients reveals a statistical trend (see Figure 20, on the left) that confirms the regularity discovered in the right-hand-tremor PD patients (see Figure 20, on the right). Note that the point cloud corresponding to the left-hand-tremor PD patients has approximately the same shape as that of the right-hand-tremor PD patient point cloud; however, the correlation coefficient is −0.6349, the first-type error probability is 0.0486, the Spearman’s rank correlation coefficient is −0.4909, and the first-type error probability in the Spearman’s nonparametric test is 0.1544. A significant correlation is not detected in the healthy subject point clouds (see Figure 20).

Let us compare the number of wave trains detected in the physiological tremor frequency band in the extensor muscle of the non-tremor hand of the PD patients with the number of wave trains detected in the Parkinsonian tremor frequency band in the extensor muscle of the tremor hand of the PD patients (see Figure 21). The abscissa axis of the scatter plot corresponds to lines 1 and 2 in Table 1. The ordinate axis corresponds to lines 5 and 6 in Table 1. We included the characteristics of the healthy subjects in the scatter plot for comparison. Each point corresponds to one subject. The PD patients are indicated by the red color; the control subjects are indicated by the green color.

The scatter plot demonstrates that the point cloud corresponding to the healthy subjects is situated along the abscissa axis. The point cloud corresponding to the PD patients has an elongated shape and is located under the healthy subject point cloud (Figure 21). Note that the point clouds can be easily separated.

Let us investigate the correlation between the number of the wave trains detected in the physiological tremor frequency band in the extensor muscle of the left non-tremor hand of the PD patients with the number of the wave trains detected in the Parkinsonian tremor frequency band in the extensor muscle of the right tremor hand of the PD patients (Figure 21, on the right). The correlation coefficient is 0.5775, the first-type error probability is 0.0493, the Spearman’s correlation coefficient is 0.5315, and the first-type error probability in the Spearman’s nonparametric test is 0.0793. Thus, a statistical trend is observed.

Let us investigate the correlation between the numbers of wave trains detected in the left-hand-tremor PD patients (Figure 21, on the left). The correlation coefficient is 0.3105 and the probability of the first-type error is 0.3826—that is, the correlation is not significant. The Spearman’s correlation coefficient is 0.4424 and the probability of the first-type error in the Spearman’s nonparametric test is 0.2042. A significant correlation is also not detected in the healthy subject point clouds (see Figure 21).

Let us compare the number of wave trains detected in the extensor muscle in the physiological tremor frequency band of the tremor hand of the PD patients with the number of wave trains detected in the extensor muscle in the physiological tremor frequency band of the non-tremor hand of the PD patients (see Figure 22). The abscissa axis corresponds to lines 3 and 4 in Table 1. The ordinate axis corresponds to lines 1 and 2 in Table 1. We included the characteristics of the healthy subjects in the scatter plot for comparison. Each point in the scatter plot corresponds to one subject. The PD patients are indicated by the red color; the control subjects are indicated by the green color.

Figure 22 demonstrates that the point clouds corresponding to the PD patients and healthy subjects can be easily separated. The healthy subject point cloud is located to the right of the PD patient point cloud.

The correlation between the number of the wave trains in the tremor and non-tremor arms of the PD patients is not significant (Figure 22). However, a statistical trend is observed in the right-hand-tremor PD patients (Figure 22, on the right). In the right-hand-tremor PD patients, the correlation coefficient is −0.4488, the first-type error probability is 0.1434, the Spearman’s correlation coefficient is −0.5455, and the first-type error probability in the Spearman’s nonparametric test is 0.0707.

In the left-hand-tremor PD patients (Figure 22, on the left), the correlation coefficient is −0.0872, the first-type error probability is 0.8107, the Spearman’s correlation coefficient is −0.1515, and the first-type error probability in the Spearman’s nonparametric test is 0.6818. No significant correlation was detected in the control subject point clouds (see Figure 22).

The analysis of the correlation between the numbers of wave trains in the physiological tremor frequency band in the tremor arms and the age of the PD patients (see Figure 23) revealed a statistically significant correlation only in the left-hand-tremor PD patients (Figure 23, on the left). The correlation coefficient is −0.7246, the first-type error probability is 0.0178, the Spearman’s correlation coefficient is −0.7356, and the first-type error probability in the Spearman’s nonparametric test is 0.0153.

Note that the correlation is not observed in the right-hand-tremor PD patients (Figure 23, on the right). The correlation coefficient is −0.0512, the first-type error probability is 0.8745, the Spearman’s correlation coefficient is −0.1399, and the first-type error probability in the Spearman’s nonparametric test is 0.6672.

No significant correlation was found between the other wave train parameters in Table 1 and age. The correlation analysis of the wave train parameters of the flexor muscle (see Table 2) revealed no significant correlation.

## 4. Discussion

The obtained results can be explained by the neurophysiological mechanisms of the tremor maintenance known nowadays.

The negative correlation between the numbers of wave trains in the frequency ranges corresponding to the Parkinsonian and physiological tremors may indicate the mutual negative influence of some neurophysiological mechanisms underlying both tremor types (Figure 20). It is known that the mechanism of physiological tremor (in the frequency range from 8 to 12 Hz) is maintained in the cerebello-thalamo-premotor-motor cortical network [41]. The Parkinsonian tremor mechanism is maintained in the cerebello-thalamic pathways [42]. Thus, the thalamus is involved in the mechanisms of both types of tremor. We can assume that the capabilities of the thalamus are limited and that both tremor mechanisms compete for the participation of the thalamus. We propose that Parkinsonian tremor is more successful than physiological one in this competition. Therefore, as Parkinsonian tremor intensifies, physiological tremor is suppressed.

We did not observe a correlation between the Parkinsonian tremor in the tremor arms and the physiological tremor in the non-tremor arms (Figure 21). However, there is a statistical trend in the right-hand-tremor PD patients. Thus, we cannot prove or deny the existence of a relation between the tremor in the tremor arms and the non-tremor arms. The relationship between the Parkinsonian tremor in the tremor arms and the physiological tremor in the non-tremor arms requires more investigations to be carried out on a larger group of corresponding patients.

No correlation was observed between the numbers of wave trains in the frequency range of the physiological tremor in the tremor arms and the numbers of the wave trains in the frequency range of physiological tremor in the non-tremor arms in the PD patients (see Figure 22). Note that there is a separation of the wave train clouds in the arms of the healthy subjects and the PD patients (Figure 22). This separation of the clouds can be explained by the fact that the physiological tremor in the healthy subjects and the physiological tremor in the PD patients have different mechanisms. It is known that the physiological tremor in healthy subjects occurs mainly due to homeostatic peripheral movements of muscles and joints to maintain posture or the movement of the limbs [43]. The resting tremor in the PD patients is associated with increased activity in the cerebello-thalamic pathways [44]. In PD patients, the increased activity of the cerebello-thalamic pathways contributes to the physiological tremor of the non-tremor arm. Thus, the Parkinsonian tremor contributes to the movement of the non-tremor arm. The cumulative tremor in the non-tremor arms of the PD patients can be named the preclinical tremor. Different characteristics of the preclinical tremor in the non-tremor limbs of the PD patients and physiological tremor in the healthy subject limbs observed during the medical examination can be used for the early diagnosis of PD.

The analysis of the correlation between the numbers of wave trains in the frequency range of the physiological tremor and the age of the PD patients revealed substantial differences between the patients with the right- and left-sided-debut of PD (Figure 23). PD usually progresses with age. Increasing the age can be considered to be a factor contributing to the progression of PD [45]. It has been suggested that the dominant and non-dominant arms may have different specializations [46,47]. Motor lateralization hypothesis [48] suggests that when the right hand is dominant, it is specialized in the predictive control of the dynamics of smooth and efficient movements. In contrast, the non-dominant left hand is specialized in the resilience to unforeseen disturbances [49,50]. Thus, there is a piece of evidence that different mechanisms control the dominant and non-dominant hands. As the non-dominant hand specializes in resistance to unforeseen disturbances, we assume that this mechanism increases with PD progression. It is possible that, in the preclinical stage, the enhancement of the stability function is more substantial than the process associated with the development of PD and the increase in the tremor.

The idea of the 3D AUC diagrams described in this paper is based on multiscale data analysis. We changed the resolution of the diagrams to look for Bonferroni correction that enables to observe statistically significant differences between the subject groups. The investigation of the multiple comparisons problem [40,51,52,53,54] is currently an important topic in biomedicine. A large number of methods have been developed to solve the multiple comparisons problem. These methods are based on the analysis of the family-wise error rate (FWE), the false discovery rate (FDR), the application of random field theory (RFT) [54,55,56], the permutation method [52], etc. Unfortunately, most of these methods do not consider the connection between the multiple comparisons problem and the problem of the multiscale analysis of biomedical data. Meanwhile, these two problems are closely related both from the point of view of the mathematical apparatus and from the point of view of the practical application of the methods. The problem with multiscale analysis is that biomedical data may contain patterns that appear on various, previously unknown scales of images/signals. It is necessary to investigate data in a certain space of scales to detect such patterns. However, different scales correspond to different numbers of multiple comparisons and, therefore, suggest a different level of statistical correction for the number of multiple comparisons. Moreover, the investigation of the data on the multiple scales implies multiple comparisons and, therefore, may require the application of additional correction for the multiple comparisons. This problem, in particular, was considered in the random field theory, and an approach to analyzing the data based on the space of scales (the scale space approach) was proposed [57]. Thus, a possible direction in the development of the method for the analysis of wave train electrical activity is the use of more accurate corrections for the multiple comparisons instead of the standard Bonferroni correction.

## 5. Conclusions

The method developed for analyzing the wave train electrical activity is a universal method for exploratory data analysis and can be applied to other types of biomedical signals [58,59,60,61,62,63,64,65,66,67,68,69,70]. In particular, we demonstrated that the statistical analysis of some characteristics of wave trains in EEG can identify features of the preclinical stage of PD [58,59,60,61]. It was found that the number of wave trains in wavelet spectrograms in the beta frequency range in first-stage PD patients was significantly reduced in comparison with the control subjects [71,72,73]. Statistically significant differences in the signals of the accelerometer [68] and EMG [63] in patients with PD, essential tremor (ET), and healthy volunteers in the 0.5–4 Hz low-frequency range were found using 2D AUC diagrams. Note that this frequency range has remained unexplored for a long time. The revealed regularities can be used for the differential diagnostics of PD and ET. The problem of the early and differential diagnostics of PD and ET by means of wave train analysis was considered in paper [69]. The source code of the Matlab program used for the analysis of EMG data has been published in GitHub [74]. The method used for the differential diagnostics of the essential tremor disease and early and first stages of Parkinson’s disease based on the wave train electrical activity analysis was patented [75].

## 6. Patents

Sushkova O.S., Morozov A.A., Gabova A.V., Karabanov A.V. Patent number RU 2741233 C1. Russian Federation. Method for differential diagnosing of essential tremor and early and first stages of Parkinson’s disease using wave train activity analysis of muscles. Published: 22.01.2021 Bul. No. 3. Application: 2020118098, 24.04.2020. Starting date of the patent validity period: 04/24/2020. Date of registration: 01/22/2021. Application date: 04/24/2020. Address for correspondence: 125009, Moscow, Mokhovaya Street, 11/7, Kotelnikov IRE RAS, Patent Department.

## Figures and Tables

**Figure 1 sensors-21-04700-f001:**
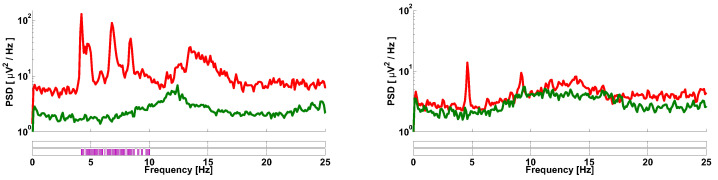
An example of the spectra of envelopes of EMG signals collected from PD patients and healthy volunteers. The spectra are computed using the standard Welch method. The Hann window was used, the window width was 10 s, and the window overlap was 7/8. The red curve indicates the PD patients. The green curve indicates the healthy volunteers. On the left, an average spectrum of the tremor right hands of twelve PD patients and an average spectrum of the right hands of ten healthy volunteers are demonstrated. On the right, an average spectrum of the non-tremor left hands of the PD patients and an average spectrum of the left hands of the healthy volunteers are demonstrated. The abscissa is the frequency. The ordinate is the power spectral density in the logarithmic scale. Two bars below the figure indicate the results of the Mann–Whitney statistical test. Statistically significant differences are indicated by the magenta color. The lower bar corresponds to the alpha level 0.05. The upper bar corresponds to the Bonferroni-corrected alpha level 0.0002. The significant differences are observed only in the tremor hands of the PD patients.

**Figure 2 sensors-21-04700-f002:**
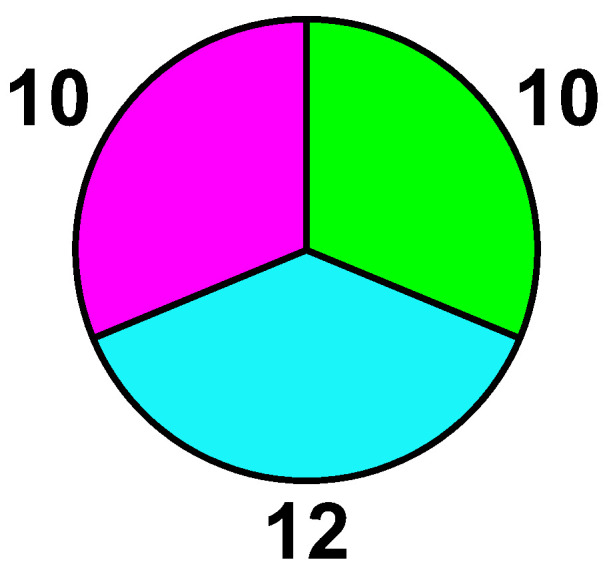
A diagram of the investigated groups of subjects. The left-hand tremor PD patients are indicated by the magenta color; the right-hand tremor PD patients are indicated by the cyan color; the healthy volunteers are indicated by the green color.

**Figure 3 sensors-21-04700-f003:**
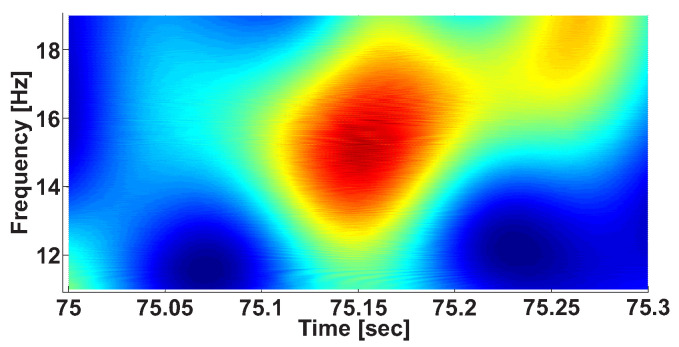
A wave train on the wavelet spectrogram of the EMG signal envelope. The abscissa axis indicates the time; the ordinate axis indicates the frequency.

**Figure 4 sensors-21-04700-f004:**
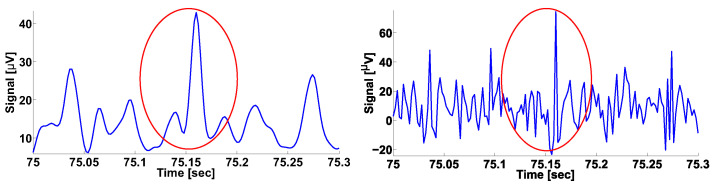
On the left is the envelope of the EMG signal considered in Figure 3. The abscissa axis indicates the time; the ordinate axis indicates the envelope of the signal in μV. On the right is the source EMG signal. The abscissa axis indicates the time; the ordinate axis indicates the amplitude of the signal in μV. The wave train is indicated by the red circle in both figures.

**Figure 5 sensors-21-04700-f005:**
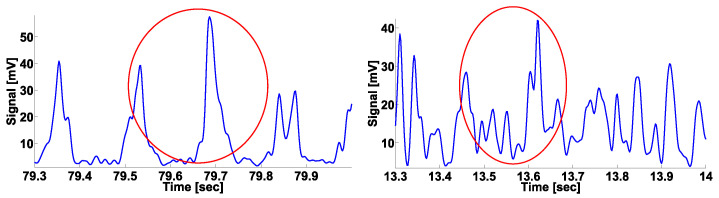
On the left: the envelope of the EMG signal in the tremor left hand of a PD patient. On the right: the envelope of the EMG signal in the left hand of a healthy volunteer. The wave trains are indicated by the red circles. The abscissa axis indicates the time; the ordinate axis indicates the envelope of the signal in μV. The wave trains are very similar. We do not try to distinguish “normal” and “abnormal” wave trains. Instead, we use a statistical analysis based on the number of detected wave trains.

**Figure 6 sensors-21-04700-f006:**
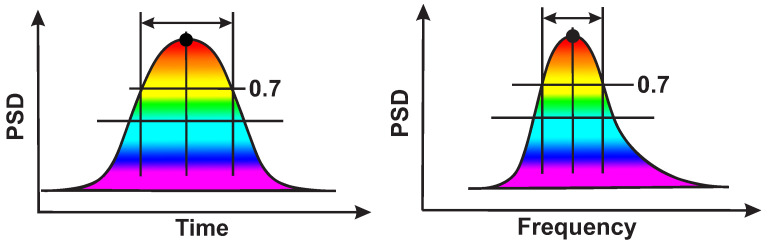
An example of a wave train spectrogram in the time–frequency domain. On the left is a time slice of the wavelet spectrogram. The abscissa is time and the ordinate is PSD. On the right is a frequency slice of the wavelet spectrogram. The abscissa is the frequency and the ordinate is PSD.

**Figure 7 sensors-21-04700-f007:**
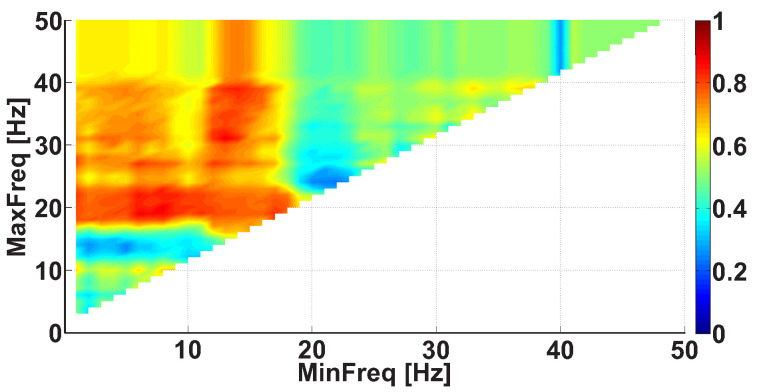
An example of the Frequency AUC diagram. The abscissa axis is the lower bound of the frequency range; the ordinate axis is the upper bound of the frequency range. Two bright red areas are observed. The first red area is situated along the abscissa axis from 0 to 18 Hz; the y-coordinate is equal to approximately 20 Hz. The second area is situated along the ordinate axis from 17 Hz and above; the x-coordinate is equal to approximately 14 Hz. The brightest point corresponds to the human physiological tremor frequency area from 8 to 20 Hz.

**Figure 8 sensors-21-04700-f008:**
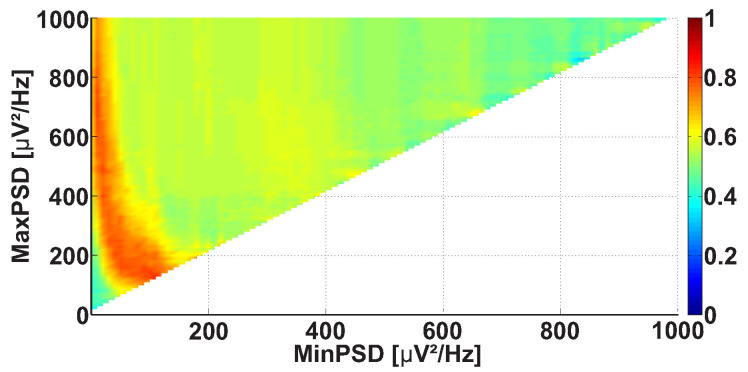
An example of the Power spectral density AUC diagram. The abscissa axis is the lower bound of the PSD range; the ordinate axis is the upper bound of the PSD range. A bright red region is observed along the ordinate axis above 70 μV${}^{2}/$ Hz.

**Figure 9 sensors-21-04700-f009:**
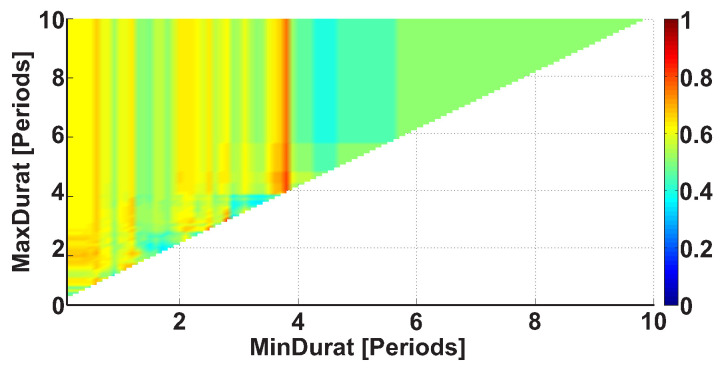
An example of the Duration AUC diagram. The abscissa axis is the lower bound of the duration range in periods; the ordinate axis is the upper bound of the duration range in periods. A narrow bright orange area is situated along the ordinate axis; the x-coordinate is equal to approximately 3.8 periods.

**Figure 10 sensors-21-04700-f010:**
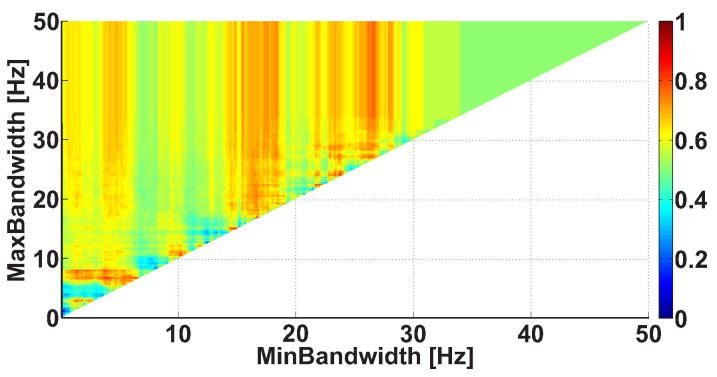
An example of the Bandwidth AUC diagram. The abscissa axis indicates the lower bound of the bandwidth; the ordinate axis indicates the upper bound of the bandwidth. The diagram demonstrates the multidirectional effects.

**Figure 11 sensors-21-04700-f011:**
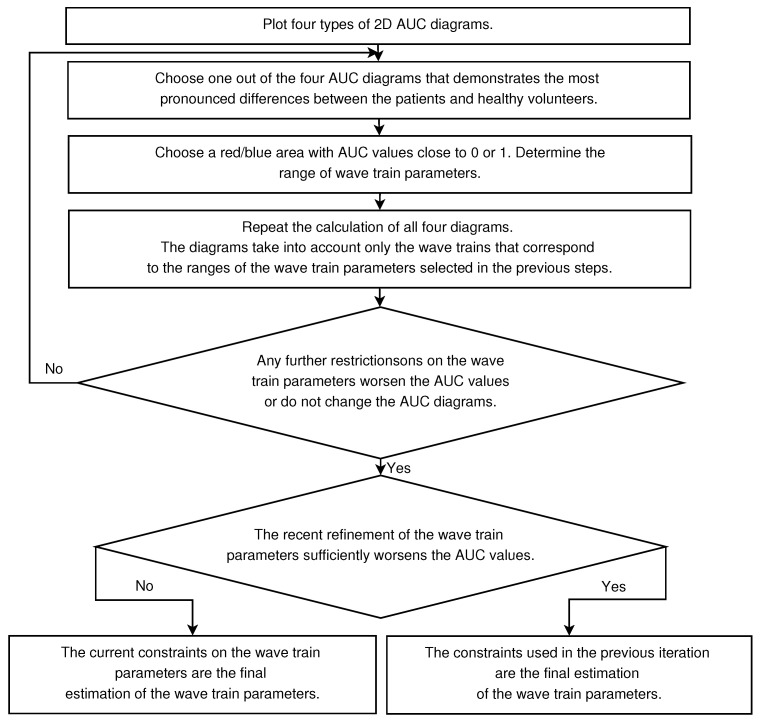
The flowchart of the method of analysis of the wave train electrical activity in EMG signals.

**Figure 12 sensors-21-04700-f012:**
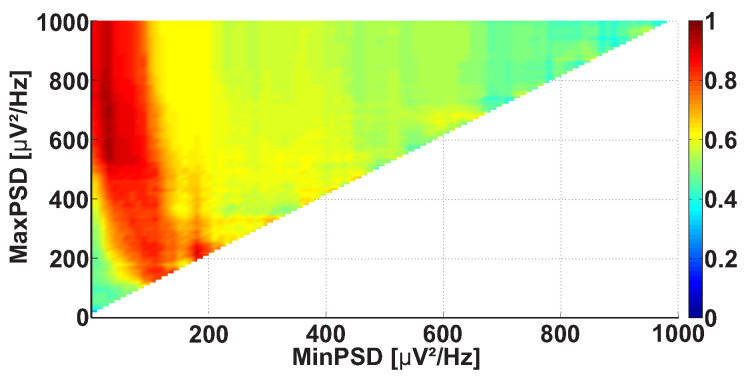
An example of the Power spectral density AUC diagram. The frequency band of the wave trains is constrained, and the frequency interval from 8 to 20 Hz is considered. The abscissa axis indicates the lower bound of the PSD range; the ordinate axis indicates the upper bound of the PSD range. A bright red area is observed along the ordinate axis; the x-coordinate is equal to approximately 30 μV${}^{2}/$ Hz. The brightest point has the following coordinates: 30 μV${}^{2}/$ Hz and 700 μV${}^{2}/$ Hz. This point corresponds to the PSD range from 30 to 700 μV${}^{2}/$ Hz.

**Figure 13 sensors-21-04700-f013:**
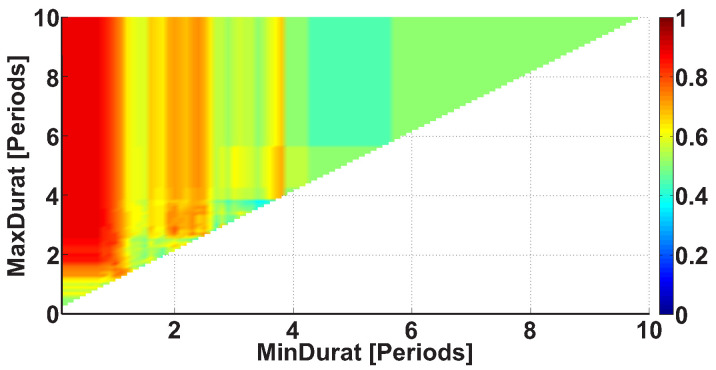
An example of the Duration AUC diagram. The frequency interval of considered wave trains is from 8 to 20 Hz. The abscissa axis indicates the lower bound of the range of durations in periods; the ordinate axis indicates the upper bound of the range of durations in periods. A bright red area is observed along the ordinate; the x-coordinate is equal to approximately 1 period. The brightest point has coordinates of 0.7 and 2.6 periods. This point corresponds to the range of durations from 0.7 to 2.6 periods.

**Figure 14 sensors-21-04700-f014:**
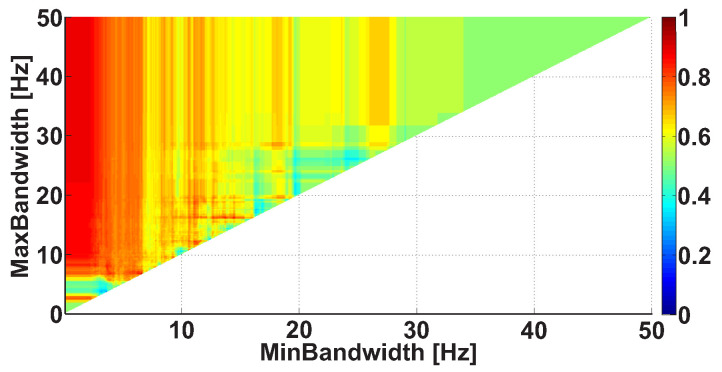
An example of the Bandwidth AUC diagram. The frequency interval of the considered wave trains is from 8 to 20 Hz. The abscissa axis indicates the lower bound of the frequency bandwidth; the ordinate axis indicates the upper bound of the frequency bandwidth. The bright red column corresponds to the frequency bandwidth values of the wave trains that are typical for PD patients.

**Figure 15 sensors-21-04700-f015:**
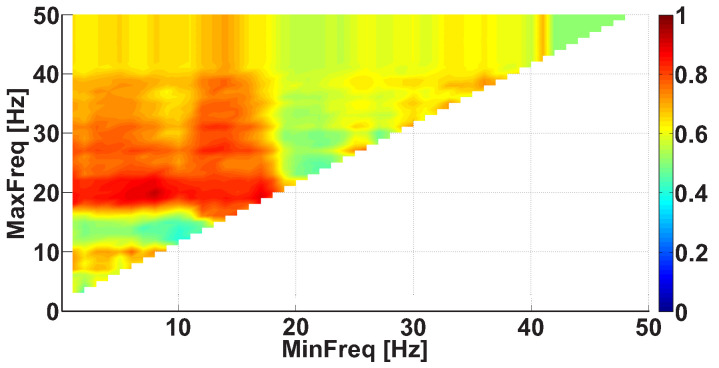
An example of the Frequency AUC diagram. The PSD of the considered wave trains is no less than 30 μV${}^{2}/$ Hz. The abscissa axis indicates the lower bound of the frequency range; the ordinate axis indicates the upper bound of the frequency range. A bright red area is observed in the frequency range along the abscissa; the y-coordinate is equal to approximately 20 Hz. The brightest red point has coordinates: 8 Hz and 20 Hz.

**Figure 16 sensors-21-04700-f016:**
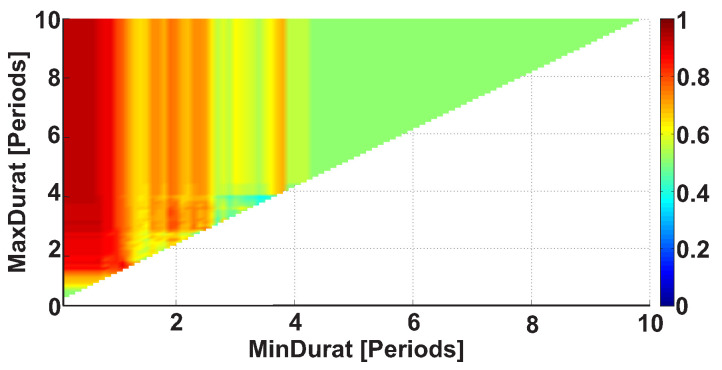
An example of the Duration AUC diagram. The following constraints are applied to the wave train parameters: a frequency from 8 to 20 Hz and a PSD no less than 30 μV${}^{2}/$ Hz. The abscissa axis indicates the lower bound of the range of the duration in periods; the ordinate axis indicates the upper bound of the range of the duration in periods. An intense red area appears inside the red region. The intense red area has the following coordinates: x-coordinates from 0 to 0.6 periods and y-coordinates from 3.6 periods and more.

**Figure 17 sensors-21-04700-f017:**
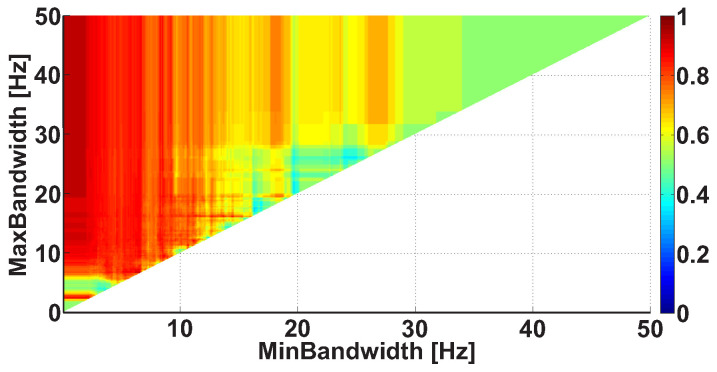
An example of the Bandwidth AUC diagram. The following constraints are applied to the wave train parameters: a frequency from 8 to 20 Hz and a PSD no less than 30 μV${}^{2}/$ Hz. The abscissa axis indicates the lower bound of the bandwidth; the ordinate axis indicates the upper bound of the bandwidth. The diagram demonstrates multidirectional effects.

**Figure 18 sensors-21-04700-f018:**
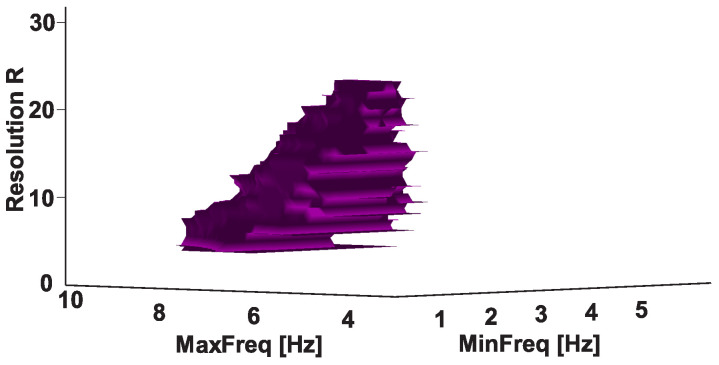
The isosurface $p\leq \alpha {}_{B}$. Left-hand-tremor PD patients; extensor muscle; frequencies from 1 to 10 Hz. The abscissa axis indicates the lower bound of the frequency range, the ordinate axis indicates the upper bound of the frequency range, and the applicate axis indicates the *R* resolution.

**Figure 19 sensors-21-04700-f019:**
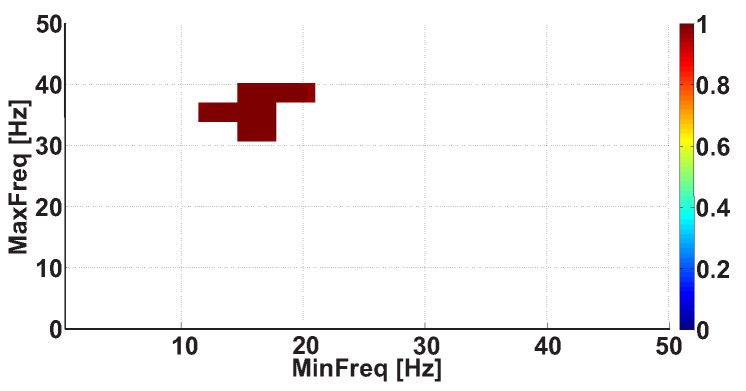
The Frequency 2D AUC diagram that corresponds to the horizontal slice of the 3D AUC diagram (see Figure 18) at the resolution *R* = 15. This Frequency 2D AUC diagram includes only the points that correspond to statistically significant differences between the groups of subjects. The abscissa axis indicates the lower bound of the frequency range; the ordinate axis indicates the upper bound of the frequency range.

**Figure 20 sensors-21-04700-f020:**
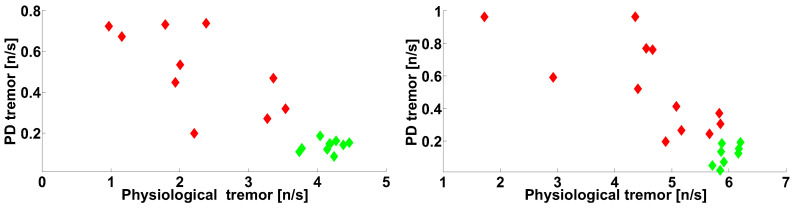
The scatter plot of the numbers of wave trains per second detected in the extensor muscle in the tremor arms of the PD patients. The abscissa axis indicates the wave train numbers corresponding to the physiological tremor; the ordinate axis indicates the Parkinsonian tremor. The PD patients are indicated by the red color. The control subjects are indicated by the green color. On the left, the left hand of the subjects are shown. On the right, the right hand of the subjects are shown.

**Figure 21 sensors-21-04700-f021:**
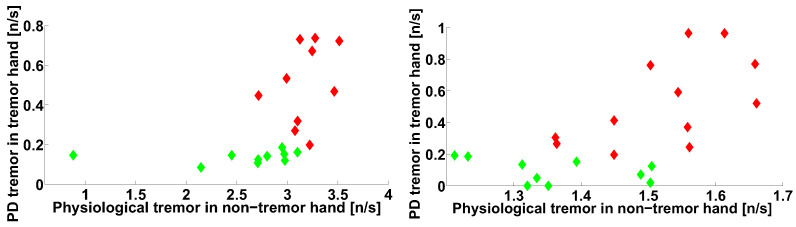
The scatter plot of the numbers of wave trains per second detected in the extensor muscle of the tremor and non-tremor arms of PD patients. The abscissa axis indicates the wave train numbers corresponding to the physiological tremor in the non-tremor hand; the ordinate axis indicates the wave train numbers corresponding to the Parkinsonian tremor in the tremor hand. The PD patients are indicated by the red color; the control subjects are indicated by the green color. On the left, the left-hand-tremor PD patients are shown. On the right, the right-hand-tremor PD patients are shown.

**Figure 22 sensors-21-04700-f022:**
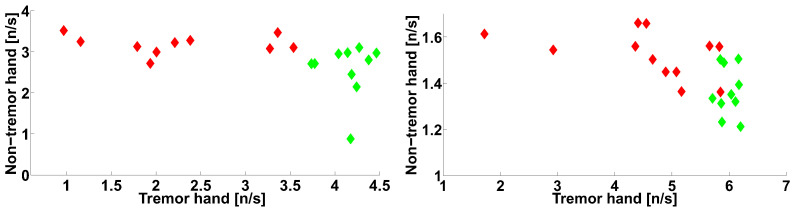
The scatter plot of the numbers of wave trains per second detected in the physiological tremor frequency band in the extensor muscle of the tremor and non-tremor arms of the PD patients. The abscissa axis indicates the wave train numbers corresponding to the physiological tremor in the tremor hand; the ordinate axis indicates the wave train numbers corresponding to the physiological tremor in the non-tremor hand. The PD patients are indicated by the red color; the control subjects are indicated by the green color. On the left, the left-hand-tremor PD patients are shown. On the right, the right-hand-tremor PD patients are shown.

**Figure 23 sensors-21-04700-f023:**
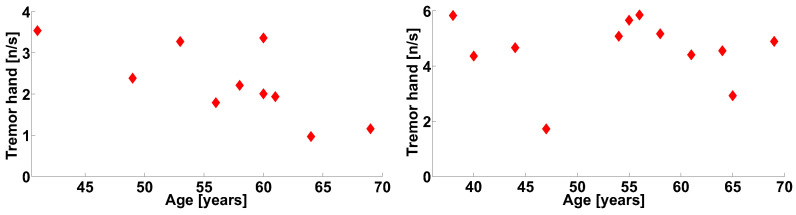
The scatter plot of the numbers of wave trains per second detected in the physiological tremor frequency band in the extensor muscle of the tremor arms of the PD patients. The scatter plot demonstrates the relation between the wave train number and the age of the patients. On the left, the left hands of the patients are shown. On the right, the right hands of the patients are shown.

**Table 1 sensors-21-04700-t001:** The characteristics of the wave trains in the extensor muscles.

Investigated Regularity	Frequency, Hz	PSD, μV${}^{2}/$ Hz	Duration, Periods	Bandwidth, Hz	AUC	*p*
A red area. The right non-tremor arm in the left-hand-tremor PD patients.	8–20	≥30	0.5–4	1–28	0.93	0.0011
A red area. The left non-tremor arm in the right-hand-tremor PD patients.	2–9	any	0.8–2.3	any	0.87	0.0033
A blue area. The left tremor arm in the left-hand-tremor PD patients.	1–50	any	≥1	≥3	0	≤0.001
A blue area. The right tremor arm in the right-hand-tremor PD patients.	6–33	any	≥0.5	≥3.5	0.02	≤0.001
A red area. The left tremor arm in the left-hand-tremor PD patients.	3–7	≥11	≥1.5	any	1	≤0.001
A red area. The right tremor arm in the right-hand-tremor PD patients.	4–8	≥103	≥1.3	any	1	≤0.0001

**Table 2 sensors-21-04700-t002:** The characteristics of the wave trains in the flexor muscles.

Investigated Regularity	Frequency, Hz	PSD, μV${}^{2}/$ Hz	Duration, Periods	Bandwidth, Hz	AUC	*p*
A red area. The right non-tremor arm in the left-hand-tremor PD patients.	5–13	0–50	any	3.1–3.8	0.92	0.0017
A red area. The left non-tremor arm in the right-hand-tremor PD patients.	2–16	any	1.4–2.1	any	0.8	0.0161
A blue area. The left tremor arm in the left-hand-tremor PD patients.	1–39	any	≥0.5	≥2.5	0	≤0.001
A blue area. The right tremor arm in the right-hand-tremor PD patients.	24–34	any	any	any	0.07	≤0.001
A red area. The left tremor arm in the left-hand-tremor PD patients.	4–7	≥4	≥1.2	any	1	≤0.001
A red area. The right tremor arm in the right-hand-tremor PD patients.	2–8	≥2	≥2.3	any	0.85	0.0037

## Data Availability

The data presented in this study are available on request from the corresponding author. The clinical data are not publicly available due to the ethical policy of the institute.

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
