# Peer review of "A Statistical Method for Exploratory Data Analysis Based on 2D and 3D Area under Curve Diagrams: Parkinson’s Disease Investigation"

_sensors, 2021, doi:10.3390/s21144700_

Round 1
Reviewer 1 Report
The comments and suggestions below are hoping to help the authors to improve the current form and strength the scientific contribution to this domain.
- There are many other methods that can perform time-frequency analysis, such as short-time Fourier transform (STFT) or Hilbert-Huang transform. The authors should address the importance and necessity of using wavelet transform on the determination of wave trains.
- In Figure 2 and Figure 3, an example EMG signals and envelope is provided. However, if the authors are intended to show that there is difference between EMG signals from patients with Parkinson’s disease and control group, an example of normal EMG (including signals during muscle contraction and relaxation) and abnormal EMG with tremor are both required to be provided.
- There are too many figures illustrating the signal processing procedures in the Method section (Figure 5 to Figure 14), not all of these figures are required to be provided in the manuscript. The authors are suggested to combine some of these figures to improve the interpretability. In addition, too many details of each procedure are described in the Method section.
- The legends of each graph (green and red) are missed in some figures (Figure 17 to Figure 20), and the units of the two axes are also missed.
- The authors claimed that high accuracy diagnosing of Parkinson’s disease can be achieved in this study. However, many of the recent studies already used wavelet-based features with artificial intelligent techniques to achieve high accuracy diagnosis of Parkinson’s disease. Please refer to the following references and elaborate your scientific contributions.
Subba, Roselene, and Akash Kumar Bhoi. "Feature Extraction and Classification Between Control and Parkinson’s Using EMG Signal." Cognitive Informatics and Soft Computing. Springer, Singapore, 2020. 45-52.
Dogan, Sengul, and Turker Tuncer. "A novel statistical decimal pattern-based surface electromyogram signal classification method using tunable q-factor wavelet transform." Soft Computing 25.2 (2021): 1085-1098.
- English editing is suggested. For example: “The method for differential diagnostics of essential tremor and early and first stages of Parkinson’s disease using …….” (line 609 – 611)
Author Response
Response to Reviewer 1 Comments
Point 1: There are many other methods that can perform time-frequency analysis, such as short-time Fourier transform (STFT) or Hilbert-Huang transform. The authors should address the importance and necessity of using wavelet transform on the determination of wave trains.
Response 1: Note that the wavelets are not a critical issue of the method of wave train analysis. Generally speaking, similar data analysis can be implemented based on the windowed Fourier transform. However, the wavelets have the following advantages: the time resolution of the wavelet changes automatically when different frequencies are investigated. Thus, the wavelets allow one to investigate wave trains simultaneously in high- and low-frequency bands. We use the Morlet wavelet because it is simple and people can easily understand the wavelet diagrams. (line 74-82)
Point 2: In Figure 2 and Figure 3, an example EMG signals and envelope is provided. However, if the authors are intended to show that there is difference between EMG signals from patients with Parkinson’s disease and control group, an example of normal EMG (including signals during muscle contraction and relaxation) and abnormal EMG with tremor are both required to be provided.
Response 2: In the framework of our method, we do not try to distinguish ``normal'' and ``abnormal'' wave trains. Instead, we use a statistical analysis based on the number of detected wave trains. A figure 4 is added that demonstrates examples of the wave trains in the tremor left hand of a PD patient and left hand of a healthy volunteer. The figure demonstrates that the wave trains are very similar. (line 157-163, Fig. 4)
Point 3: There are too many figures illustrating the signal processing procedures in the Method section (Figure 5 to Figure 14), not all of these figures are required to be provided in the manuscript. The authors are suggested to combine some of these figures to improve the interpretability. In addition, too many details of each procedure are described in the Method section.
Response 3: Yes, thank you for this comment. Yes, you are right. The method of wave train analysis can be explained in a more compact way. However, we would show how the AUC diagrams are modified during the analysis; these details can be useful from the practical point of view.
Point 4: The legends of each graph (green and red) are missed in some figures (Figure 17 to Figure 20), and the units of the two axes are also missed.
Response 4: The figures are modified. (Fig. 19-22)
Point 5: The authors claimed that high accuracy diagnosing of Parkinson’s disease can be achieved in this study. However, many of the recent studies already used wavelet-based features with artificial intelligent techniques to achieve high accuracy diagnosis of Parkinson’s disease. Please refer to the following references and elaborate your scientific contributions.
Subba, Roselene, and Akash Kumar Bhoi. "Feature Extraction and Classification Between Control and Parkinson’s Using EMG Signal." Cognitive Informatics and Soft Computing. Springer, Singapore, 2020. 45-52.
Dogan, Sengul, and Turker Tuncer. "A novel statistical decimal pattern-based surface electromyogram signal classification method using tunable q-factor wavelet transform." Soft Computing 25.2 (2021): 1085-1098.
Response 5: Our method differs from other methods based on wavelet [dogan2021novel, subba2020feature] in that the wave trains are considered and the AUC diagrams are applied. The comment is added in the text of the manuscript. (line 37, 74-82)
Point 6: English editing is suggested. For example: “The method for differential diagnostics of essential tremor and early and first stages of Parkinson’s disease using …….” (line 609 – 611)
Response 6: The text is modified. (line 627-629)
Reviewer 2 Report
The manuscript by Sushkova et al., titled, 'A Statistical Method for Exploratory Data Analysis Based on 2D and 3D AUC Diagrams: Parkinson’s Disease Investigation' aims to propose an exploratory data analysis method to analyse electroencephalogram (EEG), electromyogram (EMG), and tremorogram data acquired from Parkinsonian patients. The authors have tried to substantiate their claim that using 2D and 3D AUC diagrams provide improved detection accuracy of Parkinson's disease.
The manuscript is interesting and proposes an idea that needs to be discussed with a wider community. Hence, it can be considered to be made available to ignite further discussion. However, the current version requires substantial changes before it reaches that stage. Some of the comments that must be addressed are as follows:
- AUC is used also in the title without having defined its full form. I would suggest that its full form be used in the title.
- The authors argue that a large category of the current methods, especially, time-frequency based analysis methods fail to address the non-stationarity nature of biological signals - which is not true. Through the localisation of frequencies over time these methods have been rendering important insights into the non-stationarity of signals. Therefore, the authors must very clearly state other shortcomings of the current methods which, in their opinion, needs to be improved.
- Provide a graph depicting the population of the experimentation as described in section 2.1.
- The method should be described in more detail using flowcharts and algorithms (pseudocodes).
- The authors also must report the time complexity of the method.
- To ensure the reproduction of the results, the authors must share their code, e.g., by uploading the source code to a github repository and including the link of the same in the methodology section of the article.
- The main concern is to compare the work with other techniques. The authors are suggested to refer and cite these two review papers (10.1109/TNNLS.2018.2790388, 10.1007/s12559-020-09773-x) where machine learning-based methods have been surveyed for analysing biological signals. The authors need to compare the performance of their method with at least one of the methods presented in either of those papers.
- Remove "Prof." from the text (page 2, line 79).
- Some grammatical errors are present. Please proofread the manuscript carefully.
Author Response
Response to Reviewer 2 Comments
Point 1: AUC is used also in the title without having defined its full form. I would suggest that its full form be used in the title.
Response 1: Thank you for this comment. The title and abstract are modified. (line 1)
Point 2: The authors argue that a large category of the current methods, especially, time-frequency based analysis methods fail to address the non-stationarity nature of biological signals - which is not true. Through the localisation of frequencies over time these methods have been rendering important insights into the non-stationarity of signals. Therefore, the authors must very clearly state other shortcomings of the current methods which, in their opinion, needs to be improved.
Response 2: Yes, of course. You are right. The point is in that the standard Fourier analysis loses a big amount of information that existed in the signal but not in that another model of stationarity is used. The text is modified. Thank you for this comment. (line 13, 45)
Point 3: Provide a graph depicting the population of the experimentation as described in section 2.1.
Response 3: The diagram is added. (Fig. 1, line 117)
Point 4: The method should be described in more detail using flowcharts and algorithms (pseudocodes).
Response 4: The flowchart describing the method of wave train analysis is added. (line 289, Fig. 10)
Point 5: The authors also must report the time complexity of the method.
Response 5: The computation of wavelet spectrograms and detection of wave trains are the most time-consuming data processing steps. The processing of EMG data of the total group of subjects (32 persons) takes about 2 hours on a 2.30 GHz PC machine. The text is modified. Thank you for this question. (line 164-166)
Point 6: To ensure the reproduction of the results, the authors must share their code, e.g., by uploading the source code to a github repository and including the link of the same in the methodology section of the article.
Response 6: The reference to the GitHub repository is added. (line 626-627)
Point 7: The main concern is to compare the work with other techniques. The authors are suggested to refer and cite these two review papers (10.1109/TNNLS.2018.2790388, 10.1007/s12559-020-09773-x) where machine learning-based methods have been surveyed for analysing biological signals. The authors need to compare the performance of their method with at least one of the methods presented in either of those papers.
Response 7: Thank you for this question. Please note that it is difficult to compare the method of wave train analysis with the neural-network-based methods. The point is in that these methods address different problems. The method of wave train analysis addresses the problem of revealing basic regularities in the signal. These regularities can be useful for the recognition of neurological disease but this is not the main point. The neural-network-based methods allow to recognize efficiently neurological diseases but it is difficult to use them to reveal the regularities in the signals. Actually, these two approaches can be combined to improve each other. We have ideas on how to do it; however, this topic is out of the scope of the paper. (line 40)
Point 8: Remove "Prof." from the text (page 2, line 79).
Response 8: The text is modified. (line 88)
Point 9: Some grammatical errors are present. Please proofread the manuscript carefully.
Response 9: Yes, we did it. The text is modified.
Reviewer 3 Report
Accept after minor revision (corrections to minor methodological errors and text editing)
Author Response
Response to Reviewer 3 Comments
Point 1: Accept after minor revision (corrections to minor methodological errors and text editing).
Response 1: Thank you. The text is modified.
Round 2
Reviewer 2 Report
Thank you for addressing some of the issues raised in the previous review round. However, there is the following major concern which still remains and requires addressing.
- The research gap has not yet been well described in the revised version. The authors should make it very clear why the Area-under-Curve based method is needed in contrast to the conventional time-frequency based methods. Just mentioning that while using conventional time-frequency based methods, such as Short Time Fourier Transform, causes a large amount of information contained in the signal to be lost, is not enough and should be substantiated with necessary examples.
Author Response
Response to Reviewer 2 Comments
Point 1: The research gap has not yet been well described in the revised version. The authors should make it very clear why the Area-under-Curve based method is needed in contrast to the conventional time-frequency based methods. Just mentioning that while using conventional time-frequency based methods, such as Short Time Fourier Transform, causes a large amount of information contained in the signal to be lost, is not enough and should be substantiated with necessary examples.
Response 1: Thank you for this remark. We have added the spectra of envelopes of EMG signals collected from PD patients and healthy volunteers. The Mann-Whitney statistical test discovers statistically significant differences between the spectra of the PD patients and healthy volunteers only in the tremor hands, but not in the non-tremor hands. This example demonstrates that the conventional spectral analysis does not reveal diagnostic features of PD in the non-tremor hands of the PD patients. In this paper, we demonstrate that our method extracts much more information from the signals; in particular, statistically significant differences are detected between the EMG signals collected from the non-tremor hands of the PD patients and healthy volunteers. (lines 47-62, Fig. 1)